# Comparison of COVID-19 and Non-COVID-19 Tracheostomised Patients: Complications, Survival, and Mortality Risk Factors

**DOI:** 10.3390/jcm14020633

**Published:** 2025-01-19

**Authors:** Marta Mesalles-Ruiz, Maitane Alonso, Marc Cruellas, Martí Plana, Anna Penella, Alejandro Portillo, Víctor Daniel Gumucio, Xavier González-Compta, Manel Mañós, Julio Nogués

**Affiliations:** 1Otorhinolaryngology Department, Hospital Universitari de Bellvitge, Carrer de la Feixa Llarga, s/n, L’Hospitalet de Llobregat, 08907 Barcelona, Spain; malonso@bellvitgehospital.cat (M.A.); mcruellas@bellvitgehospital.cat (M.C.); apenella@bellvitgehospital.cat (A.P.); aportillo@bellvitgehospital.cat (A.P.); xgonzalez@bellvitgehospital.cat (X.G.-C.); mmanos@bellvitgehospital.cat (M.M.); jnogues@bellvitgehospital.cat (J.N.); 2Clinical Sciences Department, Universitat de Barcelona, Carrer de Casanova 143, 08036 Barcelona, Spain; mplanahe19@alumnes.ub.edu (M.P.); vdgumucio@gmail.com (V.D.G.); 3Intensive Care Unit, Hospital Universitari de Bellvitge, Carrer de la Feixa Llarga, s/n, L’Hospitalet de Llobregat, 08907 Barcelona, Spain; 4IDIBELL, Bellvitge Institute of Research, Gran Via de l’Hospitalet, 199, L’Hospitalet de Llobregat, 08908 Barcelona, Spain

**Keywords:** trachea, infectious diseases, airway obstruction, epidemiology, pandemics

## Abstract

**Objectives:** To compare the outcomes of tracheostomised COVID-19 patients with non-COVID-19 tracheostomised patients to identify factors influencing severity and mortality. **Methods:** A retrospective, single-centre cohort study was conducted on COVID-19 tracheostomised patients admitted from May 2020 to February 2022, compared with a cohort of non-COVID-19 tracheostomised patients. **Results:** COVID-19 tracheostomised patients had a higher mortality rate (50% vs. 27.3% in non-COVID-19 patients). Mortality risk factors in COVID-19 tracheostomised patients included female sex (HR 1.99, CI 1.09–3.61, *p* = 0.025), ischemic heart disease (HR 5.71, CI 1.59–20.53, *p* = 0.008), elevated pre-tracheostomy values of PEEP (HR 1.06, CI 1.01–1.11, *p* = 0.017) and INR (HR 1.04, CI 1.01–1.07, *p* = 0.004), and ventilatory complications (HR 8.63, CI 1.09–68.26, *p* = 0.041). No significant differences in complication rates were found based on Sars-CoV-2 infection or tracheostomy type. **Conclusions:** Tracheostomy technique did not impact complications, discharge circumstances, or mortality, supporting the safety of bedside percutaneous tracheostomies for COVID-19 patients. COVID-19 tracheostomised patients exhibited a higher mortality rate.

## 1. Introduction

The COVID-19 pandemic led to an unprecedented increase in the number of patients who presented an acute respiratory distress syndrome (ARDS) and required mechanical ventilation.

Patients with moderate hypoxemia can be treated with non-invasive mechanical ventilation (high-flow nasal cannula, CPAP, or BiPAP) on the ward, but if hypoxemia is severe, ICU admission and invasive mechanical ventilation (orotracheal intubation) are required. In patients expected to need prolonged orotracheal intubation, a tracheostomy should be considered to facilitate the weaning process, decrease respiratory fatigue, and reduce intensive care unit (ICU) stay [1,2,3,4,5].

Two types of tracheostomy technique are available: open surgical tracheostomy and percutaneous tracheostomy. Both techniques present a similar risk of complications, such as bleeding, paratracheal insertion, tracheal or esophageal laceration, pneumothorax [6]. Several predictive factors for prolonged ventilation requirements, weaning failure, and tracheostomy requirement during ICU admission have been described, including higher age, low and high body mass index (BMI), and previous cardiovascular and pneumological comorbidities [7].

During the first wave of COVID-19 in Spain (first quarter of 2020) [8], our centre’s Otorhinolaryngology Department (ENT), the Anaesthetics Department, and ICU determined that tracheostomies in COVID-19 patients would be performed by ENT specialists and nurses. A previous study conducted by our research group revealed that smoking and obesity were important risk factors for tracheostomy requirement in COVID-19 patients [9].

The risk of bleeding complications also appears to be higher in patients infected with SARS-CoV-2, likely related to the use of anticoagulant drugs to reverse the hypercoagulable state caused by SARS-CoV-2 [10]. Several studies have associated certain risk factors (obesity, smoking history, cardiometabolic disorders, and elevated inflammatory markers) with increased mortality in COVID-19 patients [9,10,11,12,13,14,15,16,17].

After the first wave, and with the experience acquired, the ICU Department started to perform bedside percutaneous tracheostomies, with ENT involvement limited to cases where percutaneous tracheostomies were contraindicated, following worldwide guideline recommendations [9,18,19,20].

This study aims to compare the evolution, complications, and mortality of COVID-19 tracheostomised patients and non-COVID-19 tracheostomised patients in our centre, analyse whether the type of tracheostomy performed can influence the evolution of these patients, and describe potential risk factors for tracheostomy in COVID-19 patients.

## 2. Materials and Methods

### 2.1. Objectives

The main objective of this study is to provide a comprehensive analysis of the evolution and outcomes of tracheostomised COVID-19 patients and compare them with non-COVID-19 tracheostomised patients.

Secondary aims are to clarify whether COVID-19 tracheostomised patients present an increased risk of haemorrhagic complications and mortality in comparison to non-COVID-19 tracheostomised patients and to analyse whether the type of tracheostomy performed influences patient outcome and prognosis.

### 2.2. Study Design and Inclusion Criteria

A single-centre retrospective observational study was conducted on all patients tracheostomised for SARS-CoV-2 infection in our centre (a university tertiary-affiliated hospital) between May 2020 and February 2022.

Data from these patients were compared with a historical cohort of non-COVID-19 ICU patients who were tracheostomised between 2018 and 2019 due to prolonged intubation for respiratory failure from any cause. These patients were admitted to the ICU for various reasons (respiratory, neurologic, sepsis, multiorgan failure, etc.), developed respiratory failure requiring prolonged mechanical ventilation, underwent unsuccessful weaning attempts, and eventually required tracheostomy.

The SARS-CoV-2 Delta variant was the only one included at the time of the study.

Underage patients and those already tracheostomised prior to ICU admission were excluded.

### 2.3. Variables

The following variables were recorded: age, sex, smoking history, body mass index (BMI), preexisting comorbidities, SARS-CoV-2 vaccination, analytic data on first day of ICU admission, anticoagulant treatment, ventilatory parameters, date of symptom onset (fever and/or dyspnea), length of hospital and ICU stay, date of tracheostomy, type of tracheostomy (surgical or percutaneous), tracheostomy bleeding (bleeding that required compressive measures, cauterization, or surgical revision of the tracheostomy site), ventilatory complications (accidental tube dislodgement, subcutaneous emphysema, mucus plug cannula obstruction), time to decannulation, subglottic stenosis, hospital discharge circumstances (home, socio-health centre, or death), 90-day survival rate, and mortality.

### 2.4. Description of the Procedures

Both procedures were performed on intubated patients under general anaesthesia.

Percutaneous tracheostomy is considered the gold standard for ICU patients. This procedure is performed bedside by intensive care physicians, eliminating the need to transfer the patient to the operating room. In our centre, we use a percutaneous dilatational tracheostomy technique. This involves inserting a guidewire into the trachea through a needle, sequentially enlarging the stoma using dilators, and placing the tracheostomy tube over the guidewire into the trachea.

Surgical tracheostomy, by contrast, is typically reserved for patients with specific anatomical or clinical challenges. These may include a short, thick neck; a history of neck surgeries or radiation; infection at the intended tracheostomy site; an unstable cervical spine; pretracheal vascularization identified via cervical ultrasound; or severe coagulopathy. The procedure is generally performed by otolaryngologists in an operating room. However, if the patient cannot be safely transferred, it can be performed at the bedside [6].

### 2.5. Ethical Aspects

In accordance with Law 14/2007, 3 July, on biomedical research [21] all data were compiled in an Excel database where the anonymity of the patients was preserved. A pseudo-anonymization process was carried out, in which the data was coded by a person external to the research team.

The study complied with the ethical guidelines of the 1975 Declaration of Helsinki [22]. It was revised and approved by the Local Ethics Committee of our centre (PR101/22). Given that this was a retrospective study and had no repercussions on the evolution or treatment of the patients and all mentioned above, it was not considered necessary to obtain informed consent.

### 2.6. Statistical Analysis

The baseline demographic and clinical profiles of the subjects for both cohorts included were described using mean and standard deviation (SD) for quantitative variables and percentages for qualitative variables. When expressing time medians, quartiles are shown in brackets. When comparing tracheostomised COVID-19 and non-COVID-19 patients: *t*-tests were applied for means comparison; Wilcoxon-Mann-Whitney tests were used for medians comparison; Chi-squared test for association between categorical variables. Cox regression was used to compare the number of days from hospital admission to death between groups of patients. When estimating the effect of tracheostomy and COVID-19 on mortality, a Cox model regression was used, adjusted for potential prognostic factors (age, sex, obesity, smoke, and heart failure). Survival curves were presented using the Kaplan-Meyer estimator and compared using the log-rank test. Survival curves at 90 days were generated to evaluate the chance of survival based on the type of tracheostomy in COVID-19 patients, non-COVID-19 patients, and for the comparison between COVID-19 and non-COVID-19 tracheostomised patients. To analyse factors associated with death, logistic regression analysis was done, adjusted for age and Sex.

The statistical significance threshold for hypothesis tests was fixed at 5%. R free software version 4.2.2 for Windows was used.

## 3. Results

### 3.1. COVID-19 Tracheostomised Patients

A total of 4015 patients infected with SARS-CoV-2 were admitted to our hospital between May 2020 and February 2022. The average hospital admission rate was 130.5 patients per month (SD 201.4) (see Figure 1). During the study period, a total of 934 patients (23.3%) required ICU admission and mechanical ventilation. Of these patients, 114 (2.8%) failed ventilatory weaning and required a tracheostomy (see flowchart in Figure 2). Of these, 21 patients (18.8%) had received the first dose of SARS-CoV-2 vaccine prior to hospital admission. The tracheostomy technique performed was percutaneous (PTrach) in 73 patients (64%) and surgical (STrach) in 41 patients (36%), with no statistically significant differences in baseline characteristics and comorbidities between both groups (see Appendix A). All patients were receiving anticoagulation treatment with heparin by ICU protocol at the time. During the follow-up, a decrease in tracheostomy requirements was observed while the percentage of SARS-CoV-2 vaccination of patients increased (see Figure 3).

The median time from the onset of the first SARS-CoV-2 symptom to hospital admission was six days [4, 9] and to ICU admission was 11 days [8, 14]. The median length of hospital admission was 66.5 days [44, 86.8], and ICU admission was 31 days [5.75, 48]. The median time of tracheostomy (from procedure to decannulation) was 46 days [35, 61]. No significant differences were observed when comparing PTrach and STrach. The periods studied are detailed in Appendix A.

Regarding ventilatory parameters previous to tracheostomy, when comparing patients with PTrach and STrach, the PEEP level on the 7th day of ICU admission (PEEP7) was the only parameter that showed statistically significant differences (*p* < 0.001), with mean values of 11 cmH_2_O (SD 2.17) for PTrach and 9.28 cmH_2_O (SD 2.29) for STrach (see Appendix A).

During the tracheostomy procedure, 4 patients (3.51%) presented with major bleeding, and one patient presented with ventilatory problems (0.88%). In the postoperative period (48 hours) we recorded 11 cases of bleeding (9.65%), 1 case of ventilatory problems (0.88%), and 4 cases of both (3.51%). Complications were stratified according to the type of tracheostomy and obesity, with no statistically significant differences observed (see Appendix A).

Regarding discharge circumstances, 16.7% (*n* = 19) were discharged home, and 33.8% (*n* = 38) required admission to a socio-health centre for recovery (Appendix A). A total of 57 patients (50%) died. The 90-day survival rate observed was similar in both groups (Figure 4A). The mortality rate was 12.67 per 1000 patients/year [8.87, 17.33] for PTrach patients and 12.68 per 1000 patients/year [7.85, 19.02] for STrach patients. When stratified by type of tracheostomy, no statistically significant differences were observed regarding survival, discharge circumstances, and mortality (detailed data in Appendix A).

The risk factors associated with mortality in the COVID-19 tracheostomised cohort were female sex (HR 1.99, CI 1.09–3.61, *p* = 0.025), ischemic heart disease (HR 5.71, CI 1.59–20.53, *p* = 0.008), elevated pre-tracheostomy values of PEEP (HR 1.06, CI 1.01–1.11, *p* = 0.017) and INR (HR 1.04, CI 1.01–1.07, *p* = 0.004), and postoperative ventilatory complications (HR 8.63, CI 1.09–68.26, *p* = 0.041) (see Appendix A).

### 3.2. Non-COVID-19 Tracheostomised Patients (Historical Cohort)

In total, 231 patients with respiratory failure were included in the non-COVID-19 tracheostomised patients’ historical cohort. Of these, 71% (*n* = 164) received a surgical tracheostomy (HSTrach) and 29% (*n* = 67) a percutaneous tracheostomy (HPTrach). All the data regarding baseline characteristics and pre-existing comorbidities is detailed in Appendix A. The mean time of hospital admission was 62 days (SD 32).

Regarding complications ratio, 19 patients presented bleeding (9.5%), 14 patients presented ventilatory complications (7%), and 9 patients presented subglottic stenosis (4.5%), with no significant differences between groups according to the type of tracheostomy (see Appendix A).

Regarding discharge circumstances, the majority of patients required admission to a social-health centre (61%, *n* = 141), and 11.7% (*n* = 27) of patients were discharged home (see Appendix A. There was a higher 90-day survival rate in patients with surgical tracheostomy compared to those with percutaneous tracheostomy (*p* = 0.0012) (see Figure 4B). 27.3% of patients died (*n* = 63) during hospital admission. The mortality rate was 9.13 for 1000 patients/year (5.79, 13.46) in HPTrach and 5.53 per 1000 patients/year (3.85, 7.6) in HSTrach (see Appendix A).

In this cohort of non-COVID-19 patients, surgical tracheostomy (HR 0.33, CI 0.17–0.61, *p* < 0.001) and heart failure (HR 3.85, CI 1.62–9.17, *p* = 0.002) were statistically significant risk factors of mortality (see Appendix A).

### 3.3. Comparison Between COVID-19 and Non-COVID-19 Tracheostomised Patients

When comparing COVID-19 and non-COVID-19 tracheostomised patients, no statistically significant differences were observed in the rate of post-tracheostomy complications (19% for COVID-19 patients vs. 18.4% for non-COVID-19 patients). See Appendix A for more detailed data.

Significant differences were observed in discharge circumstances: 50% (*n* = 57) of COVID-19 tracheostomised patients died compared to 27.3% (*n* = 63) of non-COVID-19 tracheostomised patients 33.3% (*n* = 38) of COVID-19 tracheostomised patients required nursing home care while 61% (*n* = 141) of non-COVID-19 tracheostomised patients did, and 16.7% (*n* = 19) of COVID-19 tracheostomised patients were discharged home while 11.7% (*n* = 27) of non-COVID-19 tracheostomised patients were discharged home (see Appendix A).

Moreover, the 90-day survival rate was higher in non-COVID-19 patients (68%) than in COVID-19 patients (32%) (*p* < 0.0001), suggesting that the chance of survival of non-COVID-19 patients was nearly twice as high (see Figure 4C).

The mortality rate was also significant statistically lower (*p* < 0.0001) in the non-COVID-19 tracheostomised patients (6.6 per 1000 patients/year [4.9, 8.4]) compared to COVID-19 tracheostomised cohort (12.7 per 1000 patients/year [9.6, 16.3]), detailed in Appendix A.

Overall, the risk factors related to mortality in all tracheostomised patients analysed were surgical tracheostomy (HR 0.66, CI 0.43–1.00, *p* = 0.049), SARS-CoV-2 infection (HR 2.09, CI 1.38–3.17, *p* = 0.001), increasing age every 10 years (HR 1. 21, CI 1.01–1.44, *p* = 0.033), female sex (HR 1.56, CI 1.01–2.36, *p* = 0.044), heart failure (HR 3.35, CI 1.66–6.74, *p* = 0.001), and smoking (HR 2.28, CI 1.22–4.26, *p* = 0.009) (see Appendix A).

## 4. Discussion

In this study, we analysed a cohort of tracheostomised COVID-19 patients and compared them with a historical cohort of non-COVID-19 tracheostomised patients. We observed that COVID-19 patients had a significantly higher mortality rate (50% vs. 27.3%). Additionally, we identified risk factors associated with mortality in both cohorts, including SARS-CoV-2 infection, advanced age, female sex, heart failure, and smoking. The type of tracheostomy (surgical or percutaneous) showed no significant differences in complications, mortality, or discharge circumstances in the COVID-19 cohort, although in the non-COVID-19 cohort, surgical tracheostomy was associated with higher survival, probably because historical non-COVID-19 patients with an STrach were those with a contraindication for a PTrach by guidelines (short neck, minimal neck extension, etc.) [23].

The mean length of hospital stay was similar between COVID-19 (66.5 days; SD 32) and non-COVID-19 tracheostomised patients (62 days; SD 32), consistent with findings by Alvi et al., although they reported shorter stays of 33.1 days for COVID-19 patients and 34.4 days for non-COVID-19 patients [24].

Vaccination programs for SARS-CoV-2 have proven to be highly effective in reducing the risk of severe disease, ICU admission, and mortality [25], similar to the trends observed in other well-studied older diseases like Influenza A [26]. During the follow-up, a decrease in the number of tracheostomies performed was noted as vaccination rates increased, suggesting that even a single dose may be effective at preventing severe disease. This finding supports conclusions from Liu et al.’s meta-analysis [27].

Our study also indicates that the type of tracheostomy (surgical or percutaneous) had no influence on patient outcomes, including admission time, complication rates, discharge circumstances, or mortality among COVID-19 patients. This observation aligns with the results of Ferro et al.’s meta-analysis, which, across 39 studies and 3929 COVID-19 tracheostomised patients, found no differences in mortality, decannulation times, or complication rates regarding the type of tracheostomy [3]. This is particularly important because, at the onset of the COVID-19 pandemic, there was substantial concern about how and when to perform tracheostomies safely for both patients and healthcare workers. In our centre (and surely in most others), surgical tracheostomy was considered safer to reduce complications and virus transmission but, in many cases, led to delays due to the need for specialised ENT teams. The lack of significant differences between Strach and PTrach suggests that we could potentially manage these cases in a different way. By avoiding delays and increasing the use of PTrach procedures, we could probably optimise patients’ ventilatory status more effectively, reserving STrach for specific cases.

Post-tracheostomy complications were reported in an 18.8% of patients of our cohort, with no statistically significant differences found between COVID-19 and non-COVID-19 groups. This finding is consistent with Bahk et al., who reported a 21% global post-tracheostomy complication rate [28]. Notably, bleeding complications were slightly higher in our cohort (13.2%) compared to the rates reported by Ferro et al. and Rosario et al. in their meta-analysis (7.4% and 9.2% respectively) [3,23]. Both studies described that COVID-19 patients are at an elevated risk for bleeding compared to historical cohorts of tracheostomised patients (among 6%), probably due to the prophylactic anticoagulant treatment prescribed to counteract the hypercoagulable state associated with SARS-CoV-2 infection [10,23]. This absence of differences in complications supports the previous theory about the safety of bedside PTrach, avoiding the delay in tracheostomies for COVID-19 patients.

Regarding mortality rates, the data reported varies depending on the study. Bahk et al. described similar mortality rates between tracheostomised COVID-19 and non-COVID-19 patients; however, they included non-COVID tracheostomised patients for any reason, not only for respiratory causes [28]. In contrast, Alvi et al. reported a higher 30-day mortality in tracheostomised COVID-19 patients compared to non-COVID-19 patients, with a relative risk of death of 1.37 for COVID-19 tracheostomised patients (24). This data is similar to that observed in our study, which showed a mortality rate of 50% in COVID-19 tracheostomised patients versus 27.3% in non-COVID-19 tracheostomised patients.

In COVID-19 tracheostomised patients, female sex, ischemic heart disease, elevated values of PEEP and INR pre-tracheostomy, and postoperative ventilatory complications were significantly related to mortality. These findings correlate with those described in previous studies. For example, Wang et al., in a meta-analysis of 1558 patients, identified hypertension (OR 2.3), diabetes (OR 2.47), and cardiovascular disease (OR 2.93) as significant risk factors for severe SARS-CoV-2 infection [16]. Similarly, Mitton et al. reported higher FiO2 requirements with increased 30-day mortality [29], and Grasselli et al. found that age, male sex, COPD, hypercholesterolemia, and type 2 diabetes were mortality predictors, with higher PEEP and lower PaFi ratios further elevating risk [17]. In line with these findings, our data suggest that a 1-cm H_2_O increase in PEEP level correlates with a 1.06-fold increase in mortality risk, demonstrating the importance of monitoring PEEP and PaFi ratios when considering the performance of a tracheostomy. In contrast, in our historical cohort, surgical tracheostomy was associated with higher mortality compared to percutaneous tracheostomy. This may reflect selection bias, as the sample for surgical tracheostomy was larger.

The present study has several limitations. First, as a retrospective observational single-centre study, it was exposed to a selection bias, and the data were retrieved from the health system database. In this regard, some potentially important data from the historical cohort of non-COVID-19 patients were unavailable to us, such as certain laboratory parameters (including INR) and ventilatory metrics, which could not be included in the analysis. Second, there may be a bias in comparing the cohort of COVID-19 tracheostomised patients with the historical cohort. In the historical cohort, the causes of respiratory failure were diverse and not exclusively infectious, and the circumstances under which tracheostomies were performed differed markedly from those during the pandemic. Furthermore, this study only takes into account the Delta variant, and the patients included had not completed the vaccination schedule. Further investigations need to be performed to determine applicability to newer COVID-19 variants and vaccinated populations. Despite these limitations, this study has many strengths: it compares patients who were intubated for respiratory failure either due to COVID-19 or other causes, reducing possible selection bias. Additionally, it evaluates patients who underwent tracheostomy during the pandemic under special isolation conditions, using two different techniques within the same period of study. This minimises the likelihood of bias between both groups (surgical and percutaneous COVID-19 tracheostomised patients). The study also benefits from a long follow-up period and a large sample size, further enhancing its robustness.

The study highlights the management of ICU patients requiring tracheostomy due to severe respiratory failure in the event of new COVID-19 waves or other infectious viral agents. The study underscores the need for close monitoring and protocols to prevent post-tracheostomy complications, particularly bleeding (and more caution required for patients under anticoagulant treatment). The findings may enhance individualised management and collaboration between specialists, improving clinical outcomes in tracheostomized patients.

In conclusion, the type of tracheostomy performed on COVID-19 patients did not impact patient outcomes. The tracheostomy technique used and the team performing it did not impact complication rates, hospitalisation admission, intubation, decannulation, discharge circumstances, survival, or mortality, indicating that it is safe to carry out bedside percutaneous tracheostomies on COVID-19 patients.

COVID-19 tracheostomised patients presented a higher mortality rate than non-COVID-19 patients. Mortality risk factors in COVID-19 tracheostomised patients included female sex, ischemic heart disease, elevated values of PEEP and INR pre-tracheostomy, and postoperative ventilatory complications.

## Figures and Tables

**Figure 1 jcm-14-00633-f001:**
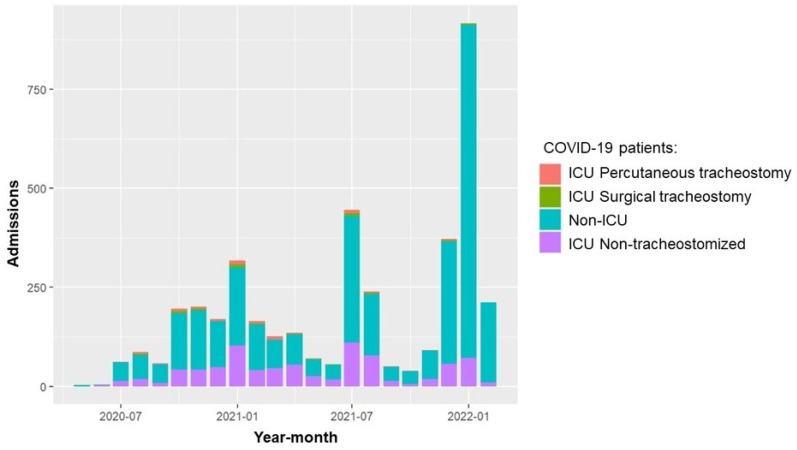
COVID-19 patients’ income progression.

**Figure 2 jcm-14-00633-f002:**
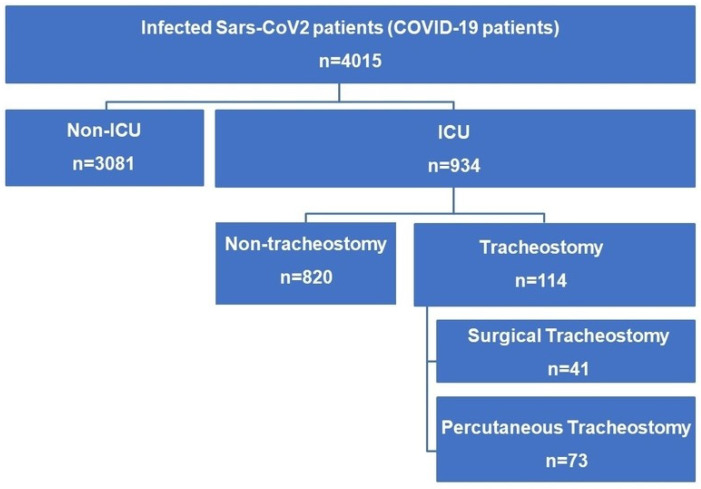
Flow chart.

**Figure 3 jcm-14-00633-f003:**
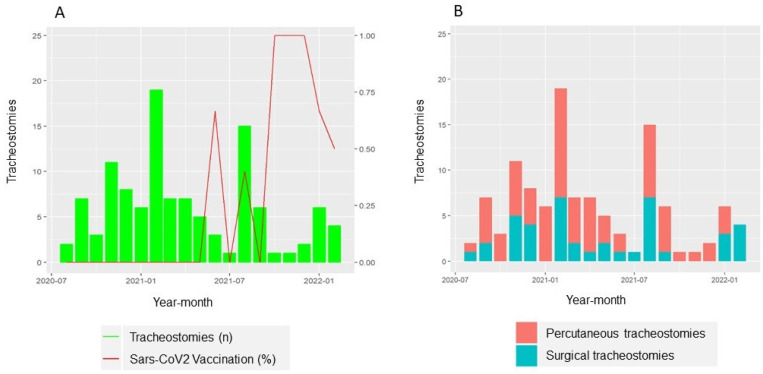
Evolution over time of the number of tracheostomies performed in COVID-19 patients. (**A**) Number of tracheostomies performed and percentage of SARS-CoV-2 vaccinated patients. (**B**) Number of tracheostomised patients divided according to the type of tracheostomy performed (percutaneous or surgical).

**Figure 4 jcm-14-00633-f004:**
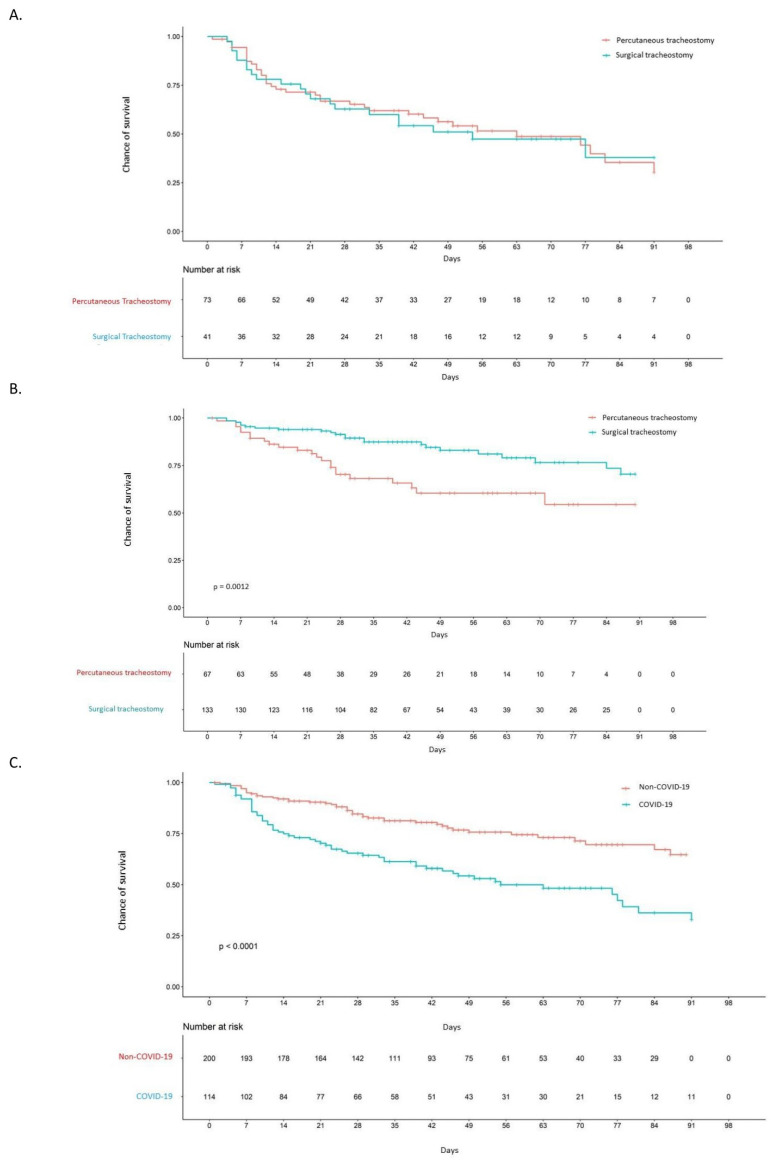
(**A**). Survival of COVID-19 tracheostomised patients (percutaneous vs. surgical tracheostomy): There are no significant differences between the two survival curves. (**B**). Survival of non-COVID-19 tracheostomsied patients (percutaneous vs. surgical tracheostomy): Statistically significant differences are observed between both survival curves, with the 90-day survival rate of patients with surgical tracheostomy higher than that of patients with percutaneous tracheostomy (*p* = 0.0012). (**C**). Survival of tracheostomised patients (COVID-19 vs. non-COVID-19): The 90-day survival rate of tracheostomised COVID-19 patients is significant lower than tracheostomized non-COVID-19 patients (*p* < 0.0001).

## Data Availability

Data is contained within the article or Appendix A.

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
