# Peer review of "Comparison of COVID-19 and Non-COVID-19 Tracheostomised Patients: Complications, Survival, and Mortality Risk Factors"

_jcm, 2025, doi:10.3390/jcm14020633_

Round 1

Reviewer 1 Report

Comments and Suggestions for Authors

Dear Authors, 

the manuscript is written in clear language, and well-structured, it adds to the current knowledge about complication rates and risk factors in COVID-19 cohorts, however, it is hard to find significant novelty or gap in the knowledge.  

The references, given the topic itself, are relevant and up-to-date. 

I have several concerns that I believe should be addressed: 

1) Please provide more details between the percutaneous vs open surgical groups - compare their outcome, what in your opinion may influence their outcome? 

2) Please unify the terminology - tracheostomy vs tracheotomy especially in the introduction - page 1 lines 38-39. 

3) Page 5 line 150 - what do you mean by "major bleeding" - was the patient on anticoagulants? In your citation number 8 the conclusion include INR as risk factor - I think it would be worth to include in your analysis. 

4) I assume that your observation time regarding survival was 90 days - it is not emphasized clearly. 

In summary, I believe this manuscript adds statistically relevant findings important to be presented scientifically, however the conclusions seem quite obvious. 

Author Response

Thank you for your expertise review and comments. We appreciate the positive comments you performed at beginning. Regarding your concerns, point by point:

1) Please provide more details between the percutaneous vs open surgical groups - compare their outcome, what in your opinion may influence their outcome? 

- Linking with other reviewer, we added a section explaining the procedures in methodology section. We explained and conclude that there were not differences between procedures. We added and extra explanation during discussion.

2) Please unify the terminology - tracheostomy vs tracheotomy especially in the introduction - page 1 lines 38-39. 

- Thank you, for detecting this mistake. We unified the terminology using only tracheostomy.

3) Page 5 line 150 - what do you mean by "major bleeding" - was the patient on anticoagulants? In your citation number 8 the conclusion include INR as risk factor - I think it would be worth to include in your analysis. 

- We added the description of major bleeding in the methodology section (in variables; line 97). For an ICU protocol all COVID-19 patients were under anticoagulation. We added at the population description (lines 158).INR was included in COVID-19 patient cohort, we described it as a risk factor associated with mortality. However, INR was not available in non-COVID-19 patients, we cannot compare INR between groups. We also added in discussion section as a limitation.

4) I assume that your observation time regarding survival was 90 days - it is not emphasized clearly. 

In summary, I believe this manuscript adds statistically relevant findings important to be presented scientifically, however the conclusions seem quite obvious. 

- We added an explanation about 90-day survival in methodology section. Moreover, we modified footnotes of figure 4 and we added an explanation about the differences in figure 4C in their corresponding sentence in results. 

Reviewer 2 Report

Comments and Suggestions for Authors

1. Introduction: line 37; to add content;  to describe in a few sentences indications for intubation and ventilation in COVID19 patients, to mention also the possibility of noninvasive ventilation modes and highflow nasal oxygenation prior intubation

2. Materials and methods: line 81, study design: to add exclusion critheria, to write some words about that topic;

- to add a new subtitle (line 82): Description of the procedures: I mean both types of tracheostomies with the type of anesthesia and personel included in the procedure

3. Discussion (line 302): objectives (hypothesis) in the conclusion should be more clearly described: the main objective and secondary aims

Author Response

Thank you for your review and comments, point by point:

  1. Introduction: line 37; to add content; to describe in a few sentences indications for intubation and ventilation in COVID19 patients, to mention also the possibility of noninvasive ventilation modes and highflow nasal oxygenation prior intubation

We added the requested content in the introduction. We explained the indications of both types of ventilations modes, also the indications for tracheostomy were included.

  1. Materials and methods: line 81, study design: to add exclusion critheria, to write some words about that topic;

We added the exclusion criteria in methodology section.

- to add a new subtitle (line 82): Description of the procedures: I mean both types of tracheostomies with the type of anesthesia and personel included in the procedure

We added this title and explained both types of tracheostomies. Thank you for the recommendation, now looks more structured and easier to read.

  1. Discussion (line 302): objectives (hypothesis) in the conclusion should be more clearly described: the main objective and secondary aims

We rewrote the conclusions to be clearer and to follow the objectives purposed in the study

Reviewer 3 Report

Comments and Suggestions for Authors

To my point of view you are comparing peas and peaches instead of focussing on one clear scientific hypothesis. There are some wonderful results - both methods are safe and complication rate are low, and, with the progress of virus mutation and specific vaccination fewer patients had to be "trached". Why do you try to "pimp" these findings with some fragile data of COVID-19 associated mortality in comparison to non COVID-19 patients?

page 5, line 150ff: specify "ventilatory problems"

page 6, Fig. 4, please add the specific legend to each Kaplan-Meier curve

page 6, Fig. 4C and page 5, line 159. Please explain the difference between the 90d survival rate of tracheotomized COVID and non-COVID patients and the significant "chance of survival" in 4C.

page 7, line 172ff: what do we expect in a hypercoagulative state? Are the groups comparable in relation to the severity of lung injury (higher PEEP in COVID-19 patients)?

Mortality is not a complication of tracheostomy, but related to the severity of disease and underlying illness and its complications. 

Please specify the origin of pulmonary failure in the non COVID group. Outcome variable could only be discussed, if pulmonary failure and consecutive tracheostomy are of a viral or bacterial origin, other than COVID-19.

Author Response

Thank you for your expertise review and comments.

We appreciate the positive comments you performed at beginning. Regarding your concerns, point by point:

To my point of view you are comparing peas and peaches instead of focussing on one clear scientific hypothesis. There are some wonderful results - both methods are safe and complication rate are low, and, with the progress of virus mutation and specific vaccination fewer patients had to be "trached". Why do you try to "pimp" these findings with some fragile data of COVID-19 associated mortality in comparison to non COVID-19 patients?

  • We discussed the absence of difference between type of tracheostomy and the decrease in the need of tracheostomy with vaccination. We would like to compare our results with other studies, focusing on the mortality. We modified the discussion section trying to focus on our strengths.

page 5, line 150ff: specify "ventilatory problems"

  • Following a similar commentary performed for Reviewer 1, we add the description of ventilatory problems in methodology section (lines 99).

page 6, Fig. 4, please add the specific legend to each Kaplan-Meier curve

  • We added a specific legend for each curve in the footnotes below the figure.

page 6, Fig. 4C and page 5, line 159. Please explain the difference between the 90d survival rate of tracheotomized COVID and non-COVID patients and the significant "chance of survival" in 4C.

  • We explained in the Results, at the corresponding sentence referring to figure 4C.

page 7, line 172ff: what do we expect in a hypercoagulative state? Are the groups comparable in relation to the severity of lung injury (higher PEEP in COVID-19 patients)?

  • In a hypercoagulative state there could be more complications, however all patients were under anticoagulation treatment for ICU protocol (added in results). Regarding the comparison of groups, you marked your commentary in line 172 (now 211), in this section there was no comparison of groups, we were studying only COVID-19 patients (percutaneous trach .vs surgical trach), for this reason we thought that the severity of lung injury would be similar. We cannot compare with non-COVID-19 patients, as an added limitations we did have not access to this data.

Mortality is not a complication of tracheostomy, but related to the severity of disease and underlying illness and its complications. 

  • We are fully agreed. We do not treated mortality as a complication of tracheostomy. We tried to separate complications and mortality during the study. We reviewed and we did not find any mistake in that topic. We are not sure if we understand this comment, could you clarify us?

Please specify the origin of pulmonary failure in the non COVID group. Outcome variable could only be discussed, if pulmonary failure and consecutive tracheostomy are of a viral or bacterial origin, other than COVID-19.

  • We specified in methodology section. Data we added a paragraph explaining that non-COVID group included all patients with respiratory failure for any reason, not only infectious disease. This was the only cohort patients that we could have access. Linking with your previous comment, we are probably comparing peaches and pees, however, the fact that there were no differences in the type of the tracheostomy, complications or admission time give more strength to the conclusion that SARS-CoV-2 infection don’t add an extra risk to the technique. We also added these thoughts at discussion and limitations.